# Using virtual reality hypnosis during stem cell transplant for patients in hematology: A protocol for a feasibility randomized study

Audrey Laurin[1,2]⊙, Floriane Rousseaux [2,3]*⊙, Valentyn Fournier[2,3,4], Jean Roy[2,4], Mathieu Landry[1,2], Richard LeBlanc[2,4], Nadia Godin[2], Caroline Arbour[5], Philippe Richebé[6], Karim Jerbi[1], Pierre Rainville[7], David Ogez[2,3]*

**1** Department of Psychology, Faculty of Arts and Sciences, University of Montreal, Montreal, Canada, **2** Maisonneuve-Rosemont Hospital Research Center, Montreal, Canada, **3** Department of Anesthesiology and Pain Medicine, Faculty of Medicine, University of Montreal, Montreal, Canada, **4** Division of Hematology, Oncology and Transplantation, Maisonneuve-Rosemont Hospital and Department of Medicine, University of Montreal, Montreal, Canada, **5** Faculty of Nursing, University of Montreal, Montreal, Canada, **6** Department of Anesthesiology and Intensive Care, Polyclinique Bordeaux Nord Aquitaine, Bordeaux, France, **7** Faculty of Dentistry, University of Montreal, Montreal, Canada

⊙ These authors contributed equally to this work.
* david.ogez@umontreal.ca (DO); floriane.rousseaux@umontreal.ca (FR)

## Abstract

### Background

Stem cell transplantation is a highly stressful treatment for hematological cancer patients, negatively impacting their quality of life. Hypnosis has proven effective in managing symptoms like pain, fatigue, and anxiety, and improving quality of life. Virtual reality (VR) is used in cancer treatments to distract from pain, and combining VR with hypnosis (VRH) could enhance the hypnotic experience. However, methodological limitations in current studies prevent clear evaluation of its effectiveness, particularly for individualized care and psychosocial interventions.

### Aims

1) To evaluate the preliminary effects of VRH in reducing anxiety, pain, and fatigue during stem cell transplant, improving quality of life post-intervention. 2) To assess intervention feasibility, including patients' experiences, satisfaction levels, and recommendations for improving the VRH tool.

### Methods

This study will involve 60 hematology patients divided into two groups: VRH and waiting list. Anxiodepressive symptoms, pain, quality of life, fatigue, absorption, dissociation, and amusement will be assessed before, immediately after the intervention, and at one- and three-month follow-ups using validated psychological scales and numeric rating scales (0–10). Semi-structured interviews will capture patient expectations,

**Data availability statement:** As a study protocol, the data are not yet available or reported in this manuscript.

**Funding:** This study was supported by the Fonds de Recherche du Québec – Santé (FRQS)/Oncopole (2023–2025, DO: 10.69777/321941 to D.O.), the Fonds de Recherche du Québec – Santé (FRQS) (2024–2025, Award BF15–341746 to F.R.), the Fonds de Recherche du Québec – Santé (FRQS) (2024–2027, DO: 10.69777/329980 to D.O.), and the Fondation de l'Hôpital Maisonneuve-Rosemont (award to A.L.). The funders had no role in study design, data collection and analysis, decision to publish, or preparation of the manuscript.

**Competing interests:** No authors have competing interests.

**Abbreviations:** VR, Virtual Reality; VRH, Virtual Reality Hypnosis; RCT, Randomized Controlled Trial; NCT, National Clinical Trial; NHL, Non-Hodgkin's Lymphoma; MANCOVA, Multivariate Analysis of Covariance; VRH, Virtual Reality Hypnosis; HADS, Hospital Anxiety and Depression Scale; BPI, Brief Pain Inventory; FACIT-BMT, Functional Assessment of Cancer Therapy – Bone Marrow Transplantation; MFI-10, Multidimensional Fatigue Inventory-10 items version; SSQ, Simulator Sickness Questionnaire; NRS, Numerical Rating Scale; CI, Confidence Interval; MANOVA, Multivariate Analysis of Variance.

satisfaction, and feedback. Data will be analyzed using MANOVA (or non-parametric alternatives) and thematic analysis.

## Discussion

The study's primary goal is to assess VRH's effectiveness compared to standard care in a feasibility randomized controlled trial (RCT). It will also provide data for improving the VRH tool (version 2.0) and assess its usability. In the long term, findings will help integrate VRH into oncology clinics, offering an innovative intervention to support patients throughout treatment.

## ClinicalTrial registration

ClinicalTrials.gov NCT06817759.

---

## 1. Introduction

### 1.1. Hematology and stem cell transplantation

Cancer patients have their quality of life severely impacted by their disease and associated treatments [1,2]. Patients with hematological cancer particularly suffer from a specific impairment since their disease affects their overall quality of life [3], increasing the risk of psychological distress [4]. Lymphomas and myeloma are common indications for autologous stem cell transplant [5]. While most stem cell transplants involve cancer patients, some are also performed for non-malignant hematologic or immune disorders. Lymphomas are classified into two main types: Hodgkin's lymphoma and non-Hodgkin's lymphoma (NHL). Multiple myeloma, the second most common hematological malignancy, is among the top ten cancer diagnoses requiring hospitalization [6]. Compared to other patients with cancer, those with lymphoma and myeloma have the greatest impairments in physical and mental health [7].

### 1.2. Quality of life

Stem cell transplantation, which includes high-dose chemotherapy followed by stem cells infusion, is recognized as one of the most stressful treatments in cancer therapy [8] due to numerous adverse effects, including pain, fatigue, nausea, weight loss [9] and uncertainty about transplant outcomes [10]. Studies have indeed shown that patients' quality of life was greatly diminished during and immediately after autologous transplantation [11], with progressive improvement during the first year of follow-up [11]. Most patients gain back a better quality of life three to five years after transplantation. Twenty-six to 36% of patients report moderate to severe depressive symptoms in the first year after transplantation, and 18% of patients experience moderate to severe anxiety in the first 100 days after transplantation [12,13]. In this context, patients suffering from depression before stem cell transplantation are more likely to have impaired functional status after [14]. The presence of depression and anxiety during the acute phase of transplantation predicts functional status, social

function, and general quality of life after transplantation [15,16]. Time spent receiving treatments represents a significant burden for patients presenting symptoms like anorexia, insomnia, and constipation, worsened by treatment adverse effects such as fatigue, nausea, and weight loss [2]. Those elements point to the necessity for developing an integrated psychological support in oncology care, especially for patients undergoing intense treatments such as stem cell transplantation [17].

### 1.3. Psychosocial interventions

The existing literature suggests that patients with hematological cancers often face anxiety, depression, and fear of recurrence. However current research tends to lack methodological rigor and long-term follow-up, limiting its ability to guide psychosocial interventions. For this population, while advancements have been made in treatment, there is a gap in high-quality studies exploring the psychosocial effects of these diseases and evaluating non-pharmacological interventions targeting them. Studies evaluating non-pharmacological interventions in oncology have shown beneficial effects on cancer adaptation and reduced levels of patients' distress and pain [18]. Among existing programs, interventions based on hypnosis techniques have been shown to be effective in the context of advanced cancers [19–22] to control common side effects such as nausea, pain, fatigue, anxiety, depressive symptoms, and improving overall quality of life [23].

### 1.4. Virtual reality hypnosis

Other interventions based on virtual reality (VR) have shown promising effects on patients' distress and acute pain [24,25]. A literature review on the usefulness of VR during cancer-related medical procedures shows that this tool is adequate to distract from pain, even for very painful procedures and particularly during chemotherapy [26]. However, the literature on VR in oncology suffers from small sample sizes, short-term evaluations, and absence of methodological robustness (e.g., self-reported, observational). Moreover, the ability of these studies to report on factors predicting the effective use of this technology (i.e., consistency, for what, whom, and when) is limited [26]. Given VR and hypnosis (VRH) are both proven to improve patient comfort, combining both techniques has become a promising field of research [27–29]. Current VR models do not offer adaptability to the patient's experience. To achieve this, it is essential to propose a tool adapted to the care context and capable of delivering beneficial effects at lower cost [26]. Given the limitations of the current literature on the development of VRH interventions, we propose a study aimed at evaluating effectiveness and feasibility of VRH, adapted to the context of stem cell transplantation.

## 2. Materials and methods

### 2.1. Objectives

This longitudinal feasibility randomized controlled trial study aims to 1) evaluate the preliminary effects of a VRH intervention in reducing symptoms of anxiety, pain, and fatigue, and in improving quality of life in patients with hematological disease during stem cell transplant, 2) Assess the feasibility of the intervention, including users's experience, satisfaction levels and recommendations for improvement of the VRH intervention and assess feasibility. Guidelines for the use of a VRH tool based on data from the literature and ergonomic recommendations were defined based on previous work [30]. The hypotheses are as follows: we believe that VRH group will decrease anxiety, pain, fatigue and increase quality of life during hospitalisation, between T0 (pre-intervention) and T3 (1 month post-intervention).

### 2.2. Ethics

This study was approved by the Ethics Committee of the Integrated Health and Social Services Centers (CISSS) and Integrated University Health and Social Services Centers (CIUSSS) of the l'île de l'Est de Montréal, Maisonneuve-Rosemont Hospital (Project n°2023–3251, date: 2023/04/25).

 

Following approval by the Maisonneuve-Rosemont Hospital Research Ethics Committee, potential participants will be invited to participate in this study. At the time of consent, participants will be advised of the confidentiality of their data, the freedom to ask any questions they wish and to stop their participation at any time without being asked for any justification and without this having any impact on their relationship with their team and patients. Participants will be provided with detailed information on the objectives and methods of the study. Participants will sign an informed consent form prior to their inclusion in the study, specifying the limits of their participation. This consent will be co-signed by the study's principal investigator. This study has many benefits for participants, but few risks are envisaged. Risks will be assessed as part of the project, and if side effects are perceived, the study will be stopped (Project n°2023–3251, date: 2023/04/25, NCT06817759).

## 2.3. Sample size and power analysis

This study will include patients undergoing stem cell transplantation in a hematology setting. The sample size for a repeated measures MANOVA (two groups, three measurement time points) was determined using an a priori power analysis with G*Power 3.1.9.7 software. The calculation was based on our primary outcome: quality of life, assessed using the FACT-BMT scale. We used an effect size $f$ of 0.25 (corresponding to a medium effect size per Cohen, 1992) [31], an alpha level of.05, and a statistical power $(1-\beta)$ of.95. This yielded a required sample size of 54 participants. Accounting for an estimated 5% attrition at each time point, the planned total sample size is 64 participants.

## 2.4. Randomization and blinding

In this randomized controlled trial, the allocation sequence will be generated using a computer-generated random number generator to ensure impartiality. Participants were randomly assigned to either the intervention or control group using a computer-generated sequence created via the web-based tool Research Randomizer (https://www.randomizer.org/) [32]. The list of random numbers was matched to participant IDs, with assignment in a 1:1 ratio, without stratification or blocking. Allocation was performed by a postdoctoral researcher not involved in data collection or analysis, ensuring allocation concealment. The postdoctoral researcher will then inform the PhD student of the randomization outcome. Both the statistician and the other investigators will remain blinded to the allocation sequence throughout the trial. The data analysts will also be blinded to the group assignments during data analysis.

## 2.5. Inclusion and exclusion criteria

Inclusion criteria will be as followed: being 18 years of age or older, suffering from hematological disease undergoing a first or second stem cell transplant, and understanding French. Exclusion criteria will be: presenting deafness, blindness, confusion, or psychiatric disorder that may impair communication according to medical records and inability to give informed consent, patients with head wounds who can not safely wear a headset, patients with a history of seizures induced by flashing lights and patients with a history of cybersickness.

## 2.6. Procedures in the hospital

Patients with hematological cancer undergoing stem cell transplantation will go through a process divided into 4 steps. The first step will be a consultation with the patient to provide information on the treatment and potential benefits/risks. In the second step, the patient will undergo a pre-transplant assessment, during which their state of health will be evaluated and the protocol adjusted. This includes blood tests, specialized consultations and a visit to the transplant unit. The third step will take place in the hospital. The patient will be admitted to a sterile room to reduce the risk of infection, and undergo high-dose chemotherapy and/or radiotherapy, aimed at destroying malignant cells and reducing the risk of graft rejection. This will be followed on day 0 by an infusion of stem cells collected from the patient or a donor via a central catheter. Immediate adverse reactions will be closely monitored. Supportive treatments (antibiotics, antivirals, blood

transfusions, analgesics, nutritional supplements) will be provided as needed, in addition to psychological and social support to deal with stress, isolation and the long-term effects of transplantation. The fourth step will take place following discharge from hospital. The transplant is considered successful when stem cells have engrafted and are producing new blood cells. After discharge, the patient will remain under close monitoring to detect potential long-term complications. In order to return to normal life, progressive rehabilitation is encouraged.

The VRH intervention will take place one week after the day of the transplant and for two weeks thereafter.

## 2.7. Procedure of the study

Recruitment will take place at Hospital Maisonneuve-Rosemont in the Hematology Clinic. Each week, the head nurse will provide the investigators with a list of potentially eligible patients. Using the *Research randomizer* program (https://www.randomizer.org/) (32), the participants will be randomly assigned to one of two groups: the experimental group receiving the VRH intervention or a control group.

We have chosen to implement a two-week protocol with a minimum of four VRH sessions for the following reasons: 1. The post-transplant period is physically and emotionally demanding. Patients often experience fatigue, nausea, and fluctuating clinical status. Our decision to select this 2-weeks window is based on clinical feasibility, to see if the intervention helps during this time lapse, but also to minimize participant burden during a vulnerable recovery window. 2. As a phase II feasibility trial, the aim is to assess feasibility, tolerability and short-term effects of the VRH intervention. The results will help inform the design of future studies with more intensive protocols, if warranted. 3. The selected time points (1- and 2-weeks post-transplant) were chosen to capture early post-transplant period, which is often characterized by elevated psychological distress. Our goal was to assess whether brief, timely intervention sessions during this critical window could offer measurable benefits. Also, patients get out of the hospital approximately 21 days post-transplant. 4. The number of sessions was also limited by equipment availability, as we had access to a limited number of headsets. This practical constraint influenced the structure of the intervention.

The first session will take place one week after transplant. During this first session, a demonstration of the use of the virtual reality headset and the *Beta Toujours Dimanche* application will be given with the help of technical data sheets (S1 Fig Technical data sheets). With the application, participants will have access to four hypnosis recordings (i.e., beach, forest, magic hand, and security protection) as well as a bank of 200 landscapes they can visit. Participants will have a first VRH session with the investigator (i.e., beach). The headset will then remain available from that day up to two weeks for use as they wish during the hospitalization. They will have the choice of using contemplative VRH (with hypnosis) or active VR (wandering through landscapes on their own). During the two weeks, three other accompanied VRH sessions are scheduled with the patient by phone or on their own to do the remaining hypnosis recordings (i.e., forest, magic hand, and security protection). Participants will be asked to complete self-evaluations at four measurement points: T0 = pre-intervention (VRH and C groups) T1 = accompanied session (VRH group), T2 = post/one month after transplantation (VRH and C groups), and T3 = follow-up/three months after transplantation (VRH and C groups), and before and after each time they use the intervention (VRH group). Interviews to gather patient expectations, satisfaction levels and recommendations (VRH group) will also be conducted at T0 and T3 respectively for patients in the VRH group (Fig 1). Auditing trial conduct will be done every week by the investigator. Participant recruitment will take place from 09/01/2025, and the follow-up period until 29/11/2025. Analysis will be completed on 31/12/2025 and the manuscript will be submitted on 10/02/2026 for publication.

## 2.8. Virtual reality hypnosis equipment

The present VRH program involves individualized patient management using hypnosis techniques, to optimize the experience and response to hypnotic suggestions for management of emotions and pain. In collaboration with Super Splendide © (www.supersplendide.com) [33], a company specializing in the development of VR applications in healthcare. Our team,

| | STUDY PERIOD | | | | |
|---|---|---|---|---|---|
| | Enrolment | Allocation | Intervention | Follow up | Close out |
| **TIMEPOINT** | -T0 | T0 | T1 | T2 | T3 |
| **ENROLMENT** | | | | | |
| Eligibility screen | X | | | | |
| Informed consent | X | | | | |
| Allocation | | X | | | |
| **INTERVENTIONS** | | | | | |
| Virtual reality hypnosis | | X | X | | |
| Waiting list | | | X | | |
| **ASSESSMENTS** | | | | | |
| Socio-demographic | | X | | | |
| Clinical information | | X | | X | |
| Expectations | | X | | | |
| Anxiety | | X | X | X | |
| Pain | | X | X | X | |
| Quality of life | | X | X | X | |
| Fatigue | | X | X | X | |
| Amusement | | | X | | |
| Relaxation | | | X | | |
| Absorption | | | X | | |
| Dissociation | | | X | | |
| Cybersickness | | | X | | |
| Satisfaction | | | | | X |
| Recommandations | | | | | X |

**Fig1. Study procedure.**

which includes two patient partners, has developed an intervention program (VRH) that combines VR and hypnosis to facilitate patients' absorption into a safe, benevolent imaginary world, in which they will be invited to live an experience that distracts them from negative emotions and pain (Beta Toujours Dimanche; Fig 2). This application was initially created for palliative and long-term care, for which over 200 volunteer hours of patient support have been carried out, with the aim of adapting developments to clinical needs. In the current project, the VRH application used with the *Meta Quest 3* virtual reality headset (Fig 2), will offer a stand-alone experience, enabling patients to take control of it during the session. It is therefore a tool that allows patients to choose the ideal timing to help them the most. The virtual experience of hypnosis

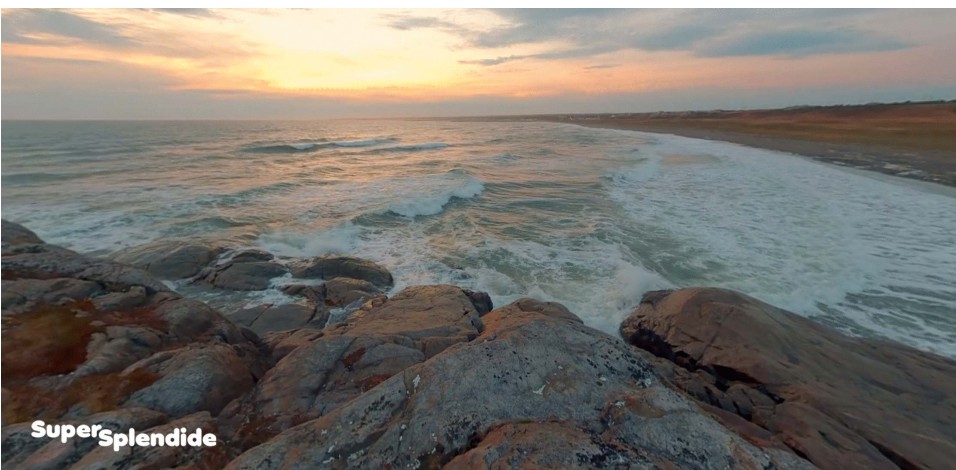

**Fig 2. Landscapes in the virtual reality program; Meta Quest 3 virtual reality headset; Beta Toujours Dimanche application logo.**

will be structured along the lines of a hypnosis session: 1- A soothing immersion will first be offered to catalyze absorption through gentle descents and calming audio-visual stimuli, 2- Reaching a safe, relaxing space, dreamily representing two soothing natural environments (beach or park, at the patient's choice), 3- Two therapeutic stress-management activities (dropping into a bag containing all the stress the patient wants to get rid of, and allowing them to be carried by an imaginary horse, which enables the patient to escape to a safe place), 4- a physical discomfort manipulation activity enabling the patient to remove the sensation of discomfort from the involved part of the body, and 5- Return to an awake state of consciousness. Moreover, some activities involve interactions for the patient (with his or her virtual environment), giving him/her a degree of agentivity towards elements usually beyond his or her control, combining suggestion with pro-action through therapy. The application allows the patient to visit a pleasant place (Fig 2) either by taking control of it or in a guided way accompanied by the hypnosis verbatim. The patient can choose between these two options, and we will document how he/she uses the application. A tablet computer is used by investigators to administer questionnaires to patients and to access the application in real time when the patient uses it.

### 2.9. Outcomes

Socio-demographic and clinical data such as biological sex, gender, age, marital status, education, conditioning, place of transplant (i.e., home or hospital), diagnosis, and treatments will be collected from patients at T0. Medication used will be taken from their medical records at T0, T2 and T3.

Primary outcomes are anxiety and pain.

Anxiodepressive symptoms will be measured with the Hospital Anxiety and Depression Scale (HADS) [34] validated in French for patients with cancer [35]. It is made up of two sub-scales containing 7 items each that determine the degree of anxiety and depression symptoms [36]. The score of each subscale is calculated by summing the score of each item, which varies between 0 and 3, so that the subscale can reach a maximum score of 21.

Pain will be assessed with the Brief Pain Inventory (BPI) [37], validated for cancer pain [38]. This scale is made of 8 items on a 10-point Likert scale assessing severity of pain, impact of pain on daily function, location of pain, pain medication and amount of pain relief in the past 24 hours or past week. It begins with a screening question about the presence of pain, accompanied by a body chart to identify painful areas and the most severe pain location. Next, the core BPI items assess pain severity and interference using numerical rating scales from 0 to 10. For pain severity, 0 represents "no pain," and 10 signifies "pain as bad as you can imagine," while for interference, 0 indicates "no interference," and 10 represents

"interferes completely." The sensory dimension (pain severity) is evaluated based on worst pain, least pain, average pain, and current pain. The reactive dimension (interference) is divided into sub-dimensions: the activity sub-dimension encompasses relationships with others, enjoyment of life, mood, and sleep, whereas the affective sub-dimension includes walking, general activity, working, and sleep. Lastly, the BPI includes items on the types of pain treatments used and their perceived effectiveness [38].

Quality of life will be assessed by the Functional Assessment of Cancer Therapy – Bone Marrow Transplantation (FACIT-BMT) [39,40]. The FACT-BMT consists of 50 items, including 27 general questions and 23 specifically tailored for blood or marrow transplant patients. Its subscales evaluate physical well-being, social/family well-being, emotional well-being, functional well-being, and "additional concerns". Each item is rated on a 0–4 scale, ranging from "not at all" to "very much," with higher scores reflecting better quality of life. The total FACT-BMT score ranges from 0 to 148.

Fatigue will be assessed with the Multidimensional Fatigue Inventory-10 items version (MFI-10) adapted for cancer [41,42]. It is scored on a 5-points Likert Scale going from 1: "do not agree at all" to 5: "totally agree".

Cybersickness symptoms related to VR immersion will be assessed with the Simulator Sickness Questionnaire validated in French [43]. It contains 16 items divided in two subscales: nausea and oculomotor, which the participant must qualify with "not at all" (score 0), "a little" (score 1), "moderately" (score 2) or "severely" (score 3). It will be assessed at T1 and three times in the following two weeks (after the use of the VRH intervention).

Usability: how patients use the application will be documented with a logbook indicating whether they chose passive or active mode and the regularity, time and duration of the use during the two weeks they have the headset in their possession... Before and after each use of the VRH intervention, patients will rate anxiety, pain, fatigue, amusement, relaxation, absorption (i.e., the tendency to become totally involved in a perceptual and imaginative experience), and dissociation (i.e., a mental separation of components of the experience) using 0–10 Numerical Rating Scales. Participants will be proposed to take part in a semi-structured interview aimed at documenting their expectations (at T0) and their satisfaction and recommendations (at T3) for improving the intervention (Table 1).

**2.9.1. Feasibility assessment.** As part of the feasibility and usability assessment of the virtual reality tool in hematology (VRH), a set of qualitative and quantitative indicators will be collected. The progression of the study and patients' motivation will be explored through semi-structured interviews, providing insight into the use of VRH during hospitalization. Dropout rates and reasons for withdrawal will be systematically recorded, along with the reported level of enjoyment and motivation to engage with the tool, the occurrence of cybersickness, and overall satisfaction. Patients' recommendations for future implementation will be also gathered. In addition, detailed monitoring of usage patterns — including preferred times, frequency, and duration of VRH sessions — enables a precise characterization of usage habits and informed potential adjustments to optimize the tool.

## 2.10. Analysis

**2.10.1. Quantitative analysis.** Continuous quantitative variables will be expressed as mean ± standard deviation, and categorical variables as headcounts and percentages. Statistical analyses will be performed using Jamovi 2.3.18 software with a threshold $\alpha = 0.05$.

Correlations will be performed on variables measured during the transplant phase (anxiety, pain, fatigue, absorption, dissociation, amusement) to measure their associations. Correlations will be made between pre- and post-intervention scores and will be estimated with Pearson's or Spearman's correlation and their 95% CI. Correlations will also be made between primary and secondary outcomes.

The assumptions of MANOVA will be tested prior to conducting the analysis using appropriate tests to assess normality of distribution, homoscedasticity, and equality of covariance matrices. If these assumptions are not met, we will employ a nonparametric alternative, specifically, the PERMANOVA (Permutatational Multivariate Analysis of Variance), which is robust to deviations from normality and appropriate for repeated measures in smaller samples. The dependent variables

**Table 1. Reported variables and measurement scales.**

| Variable | Scale |
|---|---|
| Socio-demographic | Marital status, age, biological sex, gender, education |
| Age | In years |
| Gender/biological sex | Male-female-other |
| Clinical data (medication, diagnosis, treatment, place of transplant) | Medical records and self-reported |
| Pain | Brief Pain Inventory (BPI) (37) and Numerical Rating Scale (NRS) |
| Absorption and dissociation | Numerical Rating Scale (NRS) |
| Quality of life | Functional Assessment of Cancer Therapy – Bone Marrow Transplantation (FACT-BMT) (39,40) |
| Fatigue | Multidimensional Fatigue Inventory-10 items version (MFI-10) (41,42) and Numerical Rating Scale (NRS) |
| Anxiety | Hospital Anxiety and Depression Scale (HADS) (34) and Numerical Rating Scale (NRS) |
| Depression | Hospital and Depression Scale (HADS) (34) |
| Amusement | Numerical Rating Scale (NRS) |
| Cybersickness | Simulator Sickness Questionnaire (SSQ) (43) |
| Feasibility | Semi-structured interviews |

are anxiodepressive symptoms, pain, quality of life and fatigue, and the independent variables are the different measurement times. Potential confounding variables will be considered (i.e., age, gender, socioeconomic status, absorption, dissociation, and amusement).

If conditions for parametric analysis are verified, an initial between-group comparisons at each time point, for each response variable individually, using independent samples t-tests or Wilcoxon rank-sum tests if normality is not met, will be conducted. Repeated measures ANOVA (or nonparametric equivalent) will then be conducted for each outcome separately to examine time x group interactions. Finally, a repeated measures MANOVA will be conducted to assess the overall multivariate group x time interaction across all outcomes simultaneously. This strategy allows us to (1) identify any specific differences between groups at each time point, (2) understand the temporal dynamics for each variable, and (3) explore broader multivariate patterns across outcomes.

If the MANOVA reveals significant effects, post-hoc comparisons will be conducted to explore differences between groups and across measurement times. To account for multiple comparisons while maintaining adequate statistical power, a False Discovery Rate (FDR) correction method (Benjamini-Hochberg procedure) will be applied. To confirm the robustness of findings, a Bonferroni correction (i.e., a more conservative method) will be applied too.

We expect better effects on primary outcomes, i.e., reduction of anxiety and pain levels and secondary outcomes, i.e., quality of life and fatigue for the VRH program than for control condition.

**2.10.2. Qualitative analysis.** The verbatims from the semi-structured interviews will be analyzed qualitatively using QDA Miner software.

The software enables thematic content analysis, highlighting codes in the text and graphically grouping them into themes to create a thematic tree. To identify the key themes in patients' explicit discourse, a general inductive method inspired by Braun and Clarke (2021) will be employed, aiming to uncover recurring patterns directly from the data. This data-driven approach involves deriving themes organically from the dataset [44]. The recommended process involves six main steps after preparing the raw data files (e.g., formatting and correcting transcription errors): 1) thoroughly reading the transcripts to become familiar with the data, 2) conducting line-by-line coding to generate initial codes, 3) identifying recurring themes, 4) reviewing and refining these themes, 5) defining and naming each theme, and 6) producing a report that

incorporates verbatim examples. To ensure the integrity and security of the data, a detailed plan will be followed for data entry, coding, and storage. Initially, data will be entered into a secure, password-protected database by trained personnel. To promote data quality, a double data entry process will be implemented, where two independent individuals enter the same data, and discrepancies between the entries will be flagged and resolved by a third reviewer. This will help minimize errors during the data entry process.

## 3. Discussion

Knowing that VRH interventions have respectively demonstrated efficacy in improving overall quality of life (23) as well as decreasing distress (24,25) and that VRH is a growing field of study in psychological and medical settings (27–29), the expected results of this study are an improvement in patients' quality of life and a reduction in pain, fatigue, and anxiety. These anticipated benefits align with a patient-centered approach that considers psychological well-being and personalized care. By addressing both physical and emotional distress, VRH has the potential to offer a valuable complementary tool for patients undergoing hematology treatments.

There are a few limitations to this study. As the factors predicting the effectiveness of the intervention have not been demonstrated, it is not possible to know exactly what makes it effective. Semi-structured interviews will enable us to mitigate this limitation. These are self-reported data, which leaves room for the presence of potential bias. In that sense, further studies could also include objective evaluations (e.g., physiological data). Moreover, if this study brings encouraging results, a formal RCT distinguishing each constitutive part of the intervention could be conducted to decipher which element is effective. Not using stratification or blocking in the randomization process may lead to imbalances in key baseline characteristics such as age, sex, or clinical status. This concern is particularly relevant given the small size of the study, where random variation alone may be insufficient to achieve balance between groups. Resulting imbalances in baseline characteristics could introduce confounding, potentially biasing the interpretation of group differences and diminishing statistical power. The generalizability of the findings will be limited by the single-center recruitment and the sample size. The results should be interpreted with caution, as they may not be generalizable to broader populations of different clinical settings. It will be important to conduct future multicenter studies with larger and more geographically diverse samples to better evaluate the robustness and generalizability of the VRH intervention, which will be done in the next stage of evaluation of this intervention.

Ultimately, this study will generate valuable data on the efficacy of a VRH prototype specifically designed for hematology patients. The findings will directly inform the development of an improved version (VRH 2.0), which will undergo further testing through a randomized study. By refining this intervention, we aim to contribute to the integration of VRH into standard supportive care practices for patients undergoing hematopoietic stem cell transplantation. In the long term, we hope that this technology-driven approach will be widely adopted, offering patients an innovative and accessible tool to enhance their well-being, pain symptoms, and improve their overall treatment experience.

## Supporting information

**S1 Fig. Technical data sheets.**
(DOCX)

**S1 File. CONSORT-2010-Checklist.**
(DOC)

**S2 File. Spirit checklist ok.**
(DOCX)

**S3 File. EThics all final1.**
(PDF)

 

## Acknowledgments

We express our gratitude to the Centre de gestion de la douleur and the Department of Hematology at Hôpital Maisonneuve-Rosemont for their support in this study, the dedicated physicians and nurses who contributed, the oncology clinic patients who participated, the Department of Anesthesiology, the University of Montreal, Myélome Canada, Super Splendide's chief digital artist Jean-François Malouin, our patient partners Danny Wade and Sandie Oberoi, as well as to all our collaborators for their valuable advice and interest in the project.

## Author contributions

**Conceptualization:** Floriane Rousseaux, Jean Roy, Richard LeBlanc, Mathieu Landry, Nadia Godin, Caroline Arbour, Philippe Richebé, Karim Jerbi, Pierre Rainville, David Ogez.

**Data curation:** Audrey Laurin, Floriane Rousseaux, Mathieu Landry.

**Formal analysis:** Valentyn Fournier.

**Funding acquisition:** Floriane Rousseaux, Jean Roy, Richard LeBlanc, Mathieu Landry, Caroline Arbour, Philippe Richebé, Karim Jerbi, Pierre Rainville, David Ogez.

**Investigation:** Audrey Laurin, Floriane Rousseaux, David Ogez.

**Methodology:** Audrey Laurin, Floriane Rousseaux, Jean Roy, Mathieu Landry, Philippe Richebé, Pierre Rainville, David Ogez.

**Project administration:** Jean Roy, Nadia Godin, David Ogez.

**Resources:** Jean Roy, Nadia Godin, Pierre Rainville, David Ogez.

**Software:** David Ogez.

**Supervision:** Jean Roy, Pierre Rainville, David Ogez.

**Validation:** Audrey Laurin, Floriane Rousseaux, Valentyn Fournier, Jean Roy, Richard LeBlanc, Mathieu Landry, Nadia Godin, Caroline Arbour, Philippe Richebé, Karim Jerbi, Pierre Rainville, David Ogez.

**Visualization:** Audrey Laurin, Floriane Rousseaux, Valentyn Fournier, Jean Roy, Richard LeBlanc, Mathieu Landry, Nadia Godin, Caroline Arbour, Philippe Richebé, Karim Jerbi, Pierre Rainville, David Ogez.

**Writing – original draft:** Audrey Laurin, Floriane Rousseaux.

**Writing – review & editing:** Audrey Laurin, Floriane Rousseaux.

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
