## [Decision Letter · Decision Letter 0]

16 Jul 2025

Dear Dr. Rousseaux,

Thank you for submitting your manuscript to PLOS ONE. After careful consideration, we feel that it has merit but does not fully meet PLOS ONE’s publication criteria as it currently stands. Therefore, we invite you to submit a revised version of the manuscript that addresses the points raised during the review process.

**Please provide a detailed point-by-point response along with updated version of the manuscript.**

We look forward to receiving your revised manuscript.

Kind regards,

Mohammad Mofatteh, PhD, MPH, MSc, PGCert, BSc (Hons), MB BCh (c)

Academic Editor

PLOS ONE

Journal Requirements:

2. We note that you have selected “Clinical Trial” as your article type. PLOS ONE requires that all clinical trials are registered in an appropriate registry (the WHO list of approved registries is at      https://www.who.int/clinical-trials-registry-platform/network/primary-registries" https://www.who.int/clinical-trials-registry-platform/network/primary-registries and more information on trial registration is at http://www.icmje.org/about-icmje/faqs/clinical-trials-registration/).

Please state the name of the registry and the registration number (e.g. ISRCTN or ClinicalTrials.gov) in the submission data and on the title page of your manuscript.

a) Please provide the complete date range for participant recruitment and follow-up in the methods section of your manuscript.

b) If you have not yet registered your trial in an appropriate registry, we now require you to do so and will need confirmation of the trial registry number before we can pass your paper to the next stage of review. Please include in the Methods section of your paper your reasons for not registering this study before enrolment of participants started. Please confirm that all related trials are registered by stating: “The authors confirm that all ongoing and related trials for this drug/intervention are registered”.

Please see http://journals.plos.org/plosone/s/submission-guidelines#loc-clinical-trials for our policies on clinical trials.

4. We note that the original protocol that you have uploaded as a Supporting Information file contains an institutional logo. As this logo is likely copyrighted, we ask that you please remove it from this file and upload an updated version upon resubmission.

6. Please include captions for your Supporting Information files at the end of your manuscript, and update any in-text citations to match accordingly. Please see our Supporting Information guidelines for more information: http://journals.plos.org/plosone/s/supporting-information .

Additional Editor Comments :

Please provide a detailed point-by-point response along with updated version of the manuscript.

Reviewers' comments:

Reviewer's Responses to Questions

**Comments to the Author**

1. Does the manuscript provide a valid rationale for the proposed study, with clearly identified and justified research questions?

Reviewer #1: Yes

Reviewer #2: Yes

2. Is the protocol technically sound and planned in a manner that will lead to a meaningful outcome and allow testing the stated hypotheses?

Reviewer #1: Partly

Reviewer #2: Yes

3. Is the methodology feasible and described in sufficient detail to allow the work to be replicable?

Reviewer #1: No

Reviewer #2: Yes

4. Have the authors described where all data underlying the findings will be made available when the study is complete?

Reviewer #1: Yes

Reviewer #2: Yes

5. Is the manuscript presented in an intelligible fashion and written in standard English?

*PLOS ONE*

Reviewer #1: Yes

Reviewer #2: Yes

You may also provide optional suggestions and comments to authors that they might find helpful in planning their study.

Reviewer #1: This manuscript is essentially a study protocol to conduct a longitudinal randomized controlled trial (RCT) for comparing the effectiveness of "VRH (virtual reality + hypnosis" versus "standard of care" for improving the quality of life during stem cell transplantation during hematologic treatments. The study was registered in clinicaltrials.gov, with a valid NCT number, and approved by the respective Ethics/IRB board. While the objectives and timeliness of this project appear sound and convincing, some comments appear below, following CONSORT guidelines and statistical perspectives:

(a) In Section 3, subsection 3.3, the "Randomization and Blinding" should appear as a separate subsection, with a clear focus. Details on randomization is needed (just saying computer-generated numbers doesn't suffice). In order to achieve equal participation in 2 groups, often block randomization is conducted, with some pre-decided block size. Why was that not proposed?

(b) Similarly, in Subsection 3.3, "sample size/power" should appear as a separate subsection. Although a repeated measures MANOVA will be used, it's not clear what "primary response" was used to compute the sample size? Infact, the primary response variables were mentioned much later in the manuscript.

(c) Statistical/Quantitative Analysis:

(c1) Here, it is mentioned that a MANOVA will be conducted, which is heavily based on Gaussian assumptions of the response variable. Alternatives, such as the nonparametric MANOVAs should be mentioned. Now, it is not clear what kind of nonparametric MANOVA the authors like to propose (such as, PERMANOVA, multivariate Kruskal-Wallis, etc).

(c2) Even before doing MANOVA, it's not clear why the authors initially don't like to compare the 2 groups at separate points (via 2-group t-tests/Wilcoxon rank sum tests), separately for each response variable, and then conduct a repeated measures ANOVA (and, if needed, a nonparametric counterpart) to test the group differences, separately for each response, now factoring in all time-points together. Then, they should move into the MANOVA analysis, considering multiple responses together. h

(c3) A Bonferroni correction can be too harsh; I would suggest using FDR (False Discovery rate).

(c4) It's appealing to mention the name of the software to be used to conduct the analysis.

(d) Conclusions/Discussion: This section should mention that the expected results from this study are only valid for this sample of patients recruited at a single-center hospital, and should allude to future trials with larger sample sizes and at other geographical locations to assess the effectiveness of the VRH intervention.

Reviewer #2: This is an exciting protocol that will hopefully result in a meaningful study. The N of 60 is plenty large for the goals, but I recommend a solid recruitment and headset management plan.

Some considerations in regards to the exclusion criteria: You may want to exclude patients with head wounds who can not safely wear a headset, exclude patients with a history of seizures induced by flashing lights, and exclude patients with a history of cybersickness.

I would also revisit your plan for only 2 time points for the intervention (1 week after transplant and 2 weeks thereafter). Why not prescribe this as a weekly or even daily meditation practice and assess effects further out? just two session doesn't seem nearly enough.

You mention using the Quest Pro, but your diagrams show a Quest 3. I would opt for the Quest 3 as the pro is no longer manufactured and costs much more with little benefit unless you are considering eye tracking.

Finally - consider also using the System usability Scale and Multimodal Presence Scale for your measurements after the initial session.

**Do you want your identity to be public for this peer review?** For information about this choice, including consent withdrawal, please see our Privacy Policy

Reviewer #1: No

Reviewer #2: **Yes:** Asher Marks, MD

---

## [Author Response · Author response to Decision Letter 1]

18 Sep 2025

Cover letter

Dear Editors,

Thank you for your revisions. We updated the submission. As suggested, we have resubmitted our manuscript under the Study Protocol section. Could you please confirm that it will not undergo the full peer review process again, given that it has already been reviewed by external referees?

We appreciate your clarification and look forward to your confirmation.

Best regards,

David Ogez, Pr.

Floriane Rousseaux, PhD.

Responses to editors

PONE-D-25-12703R1

Using Virtual Reality Hypnosis During Stem Cell Transplant for Patients in Hematology: A Protocol for a Randomized Study

Dr Floriane Rousseaux

Dear Dr. Rousseaux,

We've checked your submission and before we can proceed, we need you to address the following issues:

1. We notice that your revision was submitted on [Sep 10 2025], but the manuscript file in your submission's file inventory was uploaded on [Apr 14 2025]. Please upload the latest version of your revised manuscript as the main article file, with the item type 'Manuscript,' ensuring that it does not contain any tracked changes or highlighting. This will be used in the production process if your manuscript is accepted.

RESPONSE : Thank you for this observation. We removed the Apr 14 version and we add a manuscript final version in addition to the manuscript with track changes.

2. Thank you for your detailed explanation regarding the start of recruitment for your project. We regret that the peer review process has been lengthy. Unfortunately, our Registered Report Protocol format requires that recruitment begin only after the protocol is published.

Having said that, we appreciate your efforts to conduct this project with reproducibility and rigor in mind. We would be happy to consider your manuscript for publication as a Study Protocol. If you wish to have your study considered as a Study Protocol, please update the "Article Type" field in Editorial Manager accordingly. Again, my apologies for the inconvenience

RESPONSE : Thank you for considering our article as a Study Protocol. We updated the article type.

Best regards,

Floriane Rousseaux, PhD.

-------------------

Cover Letter

Responses to Editors and Reviewers

Dear Editors and Reviewers,

We would like to express our sincere gratitude to you and the reviewers for the thorough and constructive evaluation of our manuscript [PONE-D-25-12703]. We greatly appreciate the opportunity to revise our work in response to the insightful and valuable comments provided.

Please find below our detailed, point-by-point responses to each of the reviewers’ comments. We have carefully revised the manuscript to address all concerns raised. In particular, substantial modifications have been made to the Methods and Discussion sections, as well as to our statistical analysis plan, in order to improve the clarity, scientific rigor, and overall transparency of the study protocol.

We gratefully acknowledge the financial support provided by Oncopole – the cancer hub of the Fonds de recherche du Québec – Secteur Santé (FRQ-Santé) (https://oncopole.ca), the Fonds de recherche du Québec – Secteur Santé (FRQ-Santé) (https://frq.gouv.qc.ca), and specifically the FRQ-Santé “Priorité patient” Program – To Improve the Quality of Cancer Care and Service Trajectories in Québec (https://frq.gouv.qc.ca/en/program/priorite-patient-to-improve-the-quality-of-cancer-care-and-service-trajectories-in-quebec/). We also thank FRQ-Santé for the salary grant awarded to DO (n° 329980) and the FRQS Bourse Postdoctorale pour étudiants étrangers to FR (n° 2024-2025 - BF15 – 341746), as well as the Hôpital Maisonneuve-Rosemont Foundation (Fondation de l’Hôpital Maisonneuve-Rosemont) for the student grant awarded to AL (https://fondationhmr.ca/en/about-us/).

We trust that the revisions have strengthened the manuscript and we hope that it will now meet the requirements for publication in PLOS ONE.

Thank you once again for your time and consideration.

Sincerely,

Floriane Rousseaux, phD.

University of Montréal, Québec, Canada

Editors comments:

Journal Requirements:

Comment 1. Please ensure that your manuscript meets PLOS ONE's style requirements, including those for file naming. The PLOS ONE style templates can be found at

Response: We thank the editor for this reminder. We have carefully reviewed and implemented the PLOS ONE style guidelines throughout the revised manuscript.

The following adjustments were made to ensure full compliance:

- All section and subsection titles have been reformatted to match PLOS One style.

- All in-text figure references have been verified and updated.

- The titles of all figures have been revised to reflect the required formatting.

- References are now cited in brackets within the text.

- The supporting information section have been renamed and organized according to the journal’s naming conventions.

- The initials of corresponding authors have been included.

We trust that the revised submission now meets the journal’s editorial and formatting standards.

Comment 2. We note that you have selected “Clinical Trial” as your article type. PLOS ONE requires that all clinical trials are registered in an appropriate registry (the WHO list of approved registries is at https://www.who.int/clinical-trials-registry-platform/network/primary-registries" https://www.who.int/clinical-trials-registry-platform/network/primary-registries and more information on trial registration is at http://www.icmje.org/about-icmje/faqs/clinical-trials-registration/ ).

Please state the name of the registry and the registration number (e.g. ISRCTN or ClinicalTrials.gov) in the submission data and on the title page of your manuscript.

Response : We thank the editor for this important reminder. The ClinicaTrials.gov registration number as well as the link to the OSF preregistration have now been clearly indicated on the title page of the revised manuscript, and entered the appropriate fields in the submission system, in accordance with PLOS ONE’s requirements. We can now read, page 1:

- ClinicalTrial registration: NCT06817759, Protocol version 1, April 2025

- OSF : https://doi.org/10.17605/OSF.IO/Z67YA

Comment 2a. Please provide the complete date range for participant recruitment and follow-up in the methods section of your manuscript.

Response: We thank the editor for this comment. We have now added the complete date range in methods section, page 9: “Participant recruitment will take place from 09/01/2025, and the follow-up period until 29/11/2025. Analysis will be completed on 31/12/2025 and the manuscript will be submitted on 10/02/2026 for publication”

Comment 2b. If you have not yet registered your trial in an appropriate registry, we now require you to do so and will need confirmation of the trial registry number before we can pass your paper to the next stage of review. Please include in the Methods section of your paper your reasons for not registering this study before enrolment of participants started. Please confirm that all related trials are registered by stating: “The authors confirm that all ongoing and related trials for this drug/intervention are registered”.

Please see http://journals.plos.org/plosone/s/submission-guidelines#loc-clinical-trials for our policies on clinical trials.

Response: We thank the reviewer for this important comment. We confirm that the study was pre-registered on both OSF (Open Science Framework) [https://doi.org/10.17605/OSF.IO/Z67YA] and ClinicalTrials.gov: [NCT06817759]. This information is present in the title page of the manuscript. We confirm that: “The authors confirm that all ongoing and related trials for this drug/intervention are registered”.

Comment 3. We note that the grant information you provided in the ‘Funding Information’ and ‘Financial Disclosure’ sections do not match.

Response:

We gratefully acknowledge the financial support provided by Oncopole – the cancer hub of the Fonds de recherche du Québec – Secteur Santé (FRQ-Santé) (https://oncopole.ca), the Fonds de recherche du Québec – Secteur Santé (FRQ-Santé) (https://frq.gouv.qc.ca), and specifically the FRQ-Santé “Priorité patient” Program – To Improve the Quality of Cancer Care and Service Trajectories in Québec (https://frq.gouv.qc.ca/en/program/priorite-patient-to-improve-the-quality-of-cancer-care-and-service-trajectories-in-quebec/). We also thank FRQ-Santé for the salary grant awarded to DO (n° 329980) and the FRQS Bourse Postdoctorale pour étudiants étrangers to FR (n° 2024-2025 - BF15 – 341746), as well as the Hôpital Maisonneuve-Rosemont Foundation (Fondation de l’Hôpital Maisonneuve-Rosemont) for the student grant awarded to AL (https://fondationhmr.ca/en/about-us/).

Comment 3a. Response : The updated statement was changed in the cover letter.

Comment 4. We note that the original protocol that you have uploaded as a Supporting Information file contains an institutional logo. As this logo is likely copyrighted, we ask that you please remove it from this file and upload an updated version upon resubmission.

Response : We thank the editors for this reminder. Institutional logo was removed from the figure.

Comment 5. When completing the data availability statement of the submission form, you indicated that you will make your data available on acceptance. We strongly recommend all authors decide on a data sharing plan before acceptance, as the process can be lengthy and hold up publication timelines. Please note that, though access restrictions are acceptable now, your entire data will need to be made freely accessible if your manuscript is accepted for publication. This policy applies to all data except where public deposition would breach compliance with the protocol approved by your research ethics board. If you are unable to adhere to our open data policy, please kindly revise your statement to explain your reasoning and we will seek the editor's input on an exemption. Please be assured that, once you have provided your new statement, the assessment of your exemption will not hold up the peer review process.

Response : We thank the reviewer for this precision and we modificate the statement in the text. Data supporting the findings of this study are available upon reasonable request to the corresponding author. Restrictions apply to protect participant confidentiality and comply with the protocol approved by the Research Ethics Board. Anonymized and aggregated quantitative data will be shared when possible. However, qualitative data (e.g., interview transcripts) will not be made available, as there is a risk that participants could be recognized despite de-identification.

Comment 6. Please include captions for your Supporting Information files at the end of your manuscript, and update any in-text citations to match accordingly. Please see our Supporting Information guidelines for more information: http://journals.plos.org/plosone/s/supporting-information.

Response : We thank the editors. Captions were added at the end of the manuscript:

Supporting information files

S1 Fig. Technical data sheets

S2 File. Links to tutorial videos

Comment 6a. If the reviewer comments include a recommendation to cite specific previously published works, please review and evaluate these publications to determine whether they are relevant and should be cited. There is no requirement to cite these works unless the editor has indicated otherwise.

Response: We acknowledge this instruction. However, we note that the reviewers did not recommend any specific previously published works to be cited. We have therefore proceeded without adding additional citations at this stage.

Comment 7. While revising your submission, please upload your figure files to the Preflight Analysis and Conversion Engine (PACE) digital diagnostic tool, https://pacev2.apexcovantage.com/. PACE helps ensure that figures meet PLOS requirements. To use PACE, you must first register as a user. Registration is free. Then, login and navigate to the UPLOAD tab, where you will find detailed instructions on how to use the tool. If you encounter any issues or have any questions when using PACE, please email PLOS at figures@plos.org. Please note that Supporting Information files do not need this step.

Response : We thank the editors for this suggestion. We uploaded the figures to the PACE.

Comment 8. If applicable, we recommend that you deposit your laboratory protocols in protocols.io to enhance the reproducibility of your results. Protocols.io assigns your protocol its own identifier (DOI) so that it can be cited independently in the future. For instructions see:

https://journals.plos.org/plosone/s/submission-guidelines#loc-laboratory-protocols

Additionally, PLOS ONE offers an option for publishing peer-reviewed Lab Protocol articles, which describe protocols hosted on protocols.io. Read more information on sharing protocols at https://plos.org/protocols?utm_medium=editorial-email&utm_source=authorletters&utm_campaign=protocols

Response: The protocol has been uploaded on Protocols.io.

Reviewer #1:

This manuscript is essentially a study protocol to conduct a longitudinal randomized controlled trial (RCT) for comparing the effectiveness of "VRH (virtual reality + hypnosis" versus "standard of care" for improving the quality of life during stem cell transplantation during hematologic treatments. The study was registered in clinicaltrials.gov, with a valid NCT number, and approved by the respective Ethics/IRB board. While the objectives and timeliness of this project appear sound and convincing, some comments appear below, following CONSORT guidelines and statistical perspectives:

Comment 1: (a) In Section 3, subsection 3.3, the "Randomization and Blinding" should appear as a separate subsection, with a clear focus. Details on randomization is needed (just saying computer-generated numbers doesn't suffice). In order to achieve equal participation in 2 groups, often block randomization is conducted, with some pre-decided block size. Why was that not proposed?

Response: We thank the reviewer for this important comment. We have created a new subsection titled “3.4. Randomization and Blinding” to enhance clarity and focus.

3.4 Randomization and blinding

In this randomized controlled trial, the allocation sequence will be generated using a computer-generated random number generator to ensure impartiality. Participants were randomly assigned to either the intervention or control group using a computer-generated sequence created via the web-based tool Research Randomizer (https://www.randomizer.org/). The list of random numbers was matched to participant IDs, with assignment in a 1:1 ratio, without stratification or blocking. Allocation was performed by a postdoctoral researcher not involved in data collection or analysis, ensuring allocation concealment. The postdoctoral researcher will then inform the PhD student of the randomization outcome. Both the statistician and the other investigators will remain blinded to the allocation sequence throughout the trial. The data analysts will also be blinded to the group assignments during data analysis. (p.6)

We acknowledge that block randomization is commonly used to maintain group balance, especially in small samples. However, due to the preliminary nature of this phase II(b) study (ORBIT model), logistical constraints of rolling recruitment, and limited equipment availability, we opted for simple randomization to allow greater flexibility. This rationale has been added to the Methods (p. 6) and a corresponding limitation has been noted in the Discussion: “Not using stratification or blocking in th

---

## [Decision Letter · Decision Letter 1]

25 Nov 2025

Using Virtual Reality Hypnosis During Stem Cell Transplant for Patients in Hematology: A Protocol for a Randomized Study

PONE-D-25-12703R1

Dear Dr. Rousseaux,

We’re pleased to inform you that your manuscript has been judged scientifically suitable for publication and will be formally accepted for publication once it meets all outstanding technical requirements.

Kind regards,

Made Satya Nugraha Gautama, RN, M.Sc.,M.N.Sc

Academic Editor

PLOS ONE

Additional Editor Comments (optional):

Reviewers' comments:

Reviewer's Responses to Questions

**Comments to the Author**

1. Does the manuscript provide a valid rationale for the proposed study, with clearly identified and justified research questions?

Reviewer #1: Yes

Reviewer #2: Yes

2. Is the protocol technically sound and planned in a manner that will lead to a meaningful outcome and allow testing the stated hypotheses?

Reviewer #1: Yes

Reviewer #2: Yes

3. Is the methodology feasible and described in sufficient detail to allow the work to be replicable?

Reviewer #1: Yes

Reviewer #2: Yes

4. Have the authors described where all data underlying the findings will be made available when the study is complete?

Reviewer #1: Yes

Reviewer #2: Yes

5. Is the manuscript presented in an intelligible fashion and written in standard English?

Reviewer #1: Yes

Reviewer #2: Yes

You may also provide optional suggestions and comments to authors that they might find helpful in planning their study.

Reviewer #1: The authors were able to address my previous comments with great satisfaction. I have no further comments.

Reviewer #2: Thank you for the thorough and thoughtful responses to initial review. Good luck with the project and I look forward to reading more about it in the future.

**Do you want your identity to be public for this peer review?** For information about this choice, including consent withdrawal, please see our Privacy Policy

Reviewer #1: No

Reviewer #2: No

---

## [Editor Report · Acceptance letter]

PONE-D-25-12703R1

PLOS One

Dear Dr. Rousseaux,

I'm pleased to inform you that your manuscript has been deemed suitable for publication in PLOS One. Congratulations! Your manuscript is now being handed over to our production team.

Kind regards,

on behalf of

Mr. Made Satya Nugraha Gautama

Academic Editor

PLOS One